# Postpartum Body Condition Score (BCS) and Lactation Stage (30 and 60 Days) Affecting Essential Fatty Acids (EFA) and Milk Quality of Najdi Sheep

**DOI:** 10.3390/vetsci10090552

**Published:** 2023-09-02

**Authors:** Abdulkareem M. Matar, Riyadh S. Aljummah

**Affiliations:** Department of Animal Production, College of Food and Agriculture Sciences, King Saud University, P.O. Box 2460, Riyadh 11451, Saudi Arabia; rjummah@ksu.edu.sa

**Keywords:** body condition score, lactation stage, fatty acid profile, milk composition, Najdi breed

## Abstract

**Simple Summary:**

Through agricultural management, with appropriate knowledge and application, it is possible to identify the best conditions for the production of high-quality dairy products. The physical deterioration of ewes in the first week after birth is not due solely to the onset of milk production but also to a rapid increase in milk production. However, milk production in sheep peaks four weeks after parturition. In this study, both the composition and fatty acid profile of milk were analyzed to determine the correlation and effects of the body condition of Najdi ewes after parturition and their lactation stage at days 30 and 60. The BCS after parturition was positively correlated with milk fat (0.049) and margaric acid C17:0 (0.023), while it was negatively correlated with linoleic acid(AL-C18:2) (−0.002), conjugated linoleic acids (CLA-C18:2) (−0.03), and arachidonic acid C20:4 (−0.01). Milk from ewes with a BCS of 2.5 had high levels of UFA and MUFA at day 60 of lactation compared to other ewes.

**Abstract:**

Body condition scoring (BCS) can be used to assess the energy reserves of sheep during feeding, production, and weaning. The aim of this study was to evaluate the influence of BCS after parturition in stages of lactation (30 and 60 days) on the milk quality of Najdi ewes. The ewes were milked in the morning after their lambs had been isolated. Milk composition and fatty acid profiles (FA) were analyzed at 30 and 60 days of lactation after assessment of the sheep’s body condition. The sheep were classified into the following body conditions: 2.5, 3.0, 3.5, and 4.0. Sheep milk contained significant (*p* < 0.05) levels of protein at a BCS of 3.5 and on day 60 of lactation. The ewes with a BCS of 2.5 had a high milk content (*p* < 0.05) of unsaturated fatty acids (USFA), monounsaturated fatty acid (MUFA), oleic acid (OA), vaccenic acid (VA), and LA at day 60 of lactation. This result shows that the ewes with a BCS of 2.5 were able to produce high-quality milk, and 60 days of lactation was the preferred time for producing good milk and tasty and healthy dairy products.

## 1. Introduction

Over the past three decades, there has been research activity on dairy animals aiming at more efficient production and better-quality products. It is not just a question of high milk yield during lactation, but also of efficient herd management in order to obtain high-quality products. Unfortunately, although the goal has always been to increase milk production, this has side effects that further weaken animals through postpartum diseases, stress, and declining milk quality. Milk quality is influenced by various factors such as nutrition, genetics, lactation stage, animal physiology, management (udder health, milking hygiene, and BCS), etc. [1]. Therefore, the body condition score system is an important tool in dairy farming management. With appropriate knowledge and application, it is possible to determine the best conditions for high-quality production.

Body condition scoring is an easy-to-use indicator for determining the energy reserves of sheep by assessing the amount of fat coverage and muscle on the animal’s body, often using a numerical scoring system. BCS can be influenced by factors such as diet, breed, lactation stage, and general management practices [2].

Maintaining an appropriate body condition in dairy sheep is crucial for optimal milk production and milk quality. On the other hand, the loss of physical condition after parturition due to the onset of lactation requires the mobilization of physical reserves as products of negative energy balance (NEB) can affect milk production [2].

In dairy sheep, lactating ewes typically reach their maximum milk production (peak of lactation) at 3–4 weeks after lambing and produce 75% of their total milk yield in the first 8 weeks of lactation [3]. Peak lactation is determined based on milk production traits (estimation of lactation stage, selection of animals based on their performance curve, lactation persistence, etc.) or for technical applications (variations in reaching peak lactation depending on time of birth, feeding, body condition, etc.) [4]. Hynes et al. [5] points out that lipid composition in sheep is one of the most important components in terms of high-quality and nutritional milk. Lipid and FA content affect cheese yield and firmness, as well as the color and flavor of the resulting dairy products and have been shown to have health benefits including omega-3 fatty acids such as conjugated linoleic acid (CLA) [6,7].

Milk components and fatty acids absorbed from the blood by the udder make up about 60% of milk fat and are incorporated into triacylglycerols [8]. The synthesis of fatty acids from body fat stores is relevant in NEB situations when the fatty acids are esterified and bound to albumin, which is transported in the blood and available in the udder [9]. The mobilization of the body fat leads to the synthesis of palmitic acid (C16:0), stearic acid (C18:0), and oleic acid (C18:1 cis 9) in the mammary gland [8], as well as relevant amounts of branched-chain fatty acids with 17 carbon atoms [10].

During the last 2 to 4 weeks of pregnancy, there is a significant increase in energy requirements with a simultaneous decrease in dry matter intake; these two circumstances are responsible for the NEB that sets in a few weeks before delivery [11]. This implies a mobilization of body fat and a consequent increase in blood non-esterified fatty acids (NEFA), especially 2 to 3 days before delivery, reaching its maximum at the time of delivery [9]. On the other hand, [11] indicates that excessive loss of body condition is associated with a reduction in fertility and milk production.

Based on these points and in the context of the occurrence of NEB in sheep and its consequences for their milk quality and FA profile, we conducted an experiment on the body condition of Najdi ewes after parturition and its effects on milk composition during the lactation stage. The aim of this study was to evaluate the influence of the body condition and lactation stage at days 30 and 60 on milk composition and the fatty acid profile of the milk of Najdi ewes.

## 2. Materials and Methods

### 2.1. Management of Animals

A total of 126 multiparous ewes aged 2.5 to 4 years with a single lamb were randomly selected for this study from the Al-Khalidiyyah sheep farm in Riyadh under the supervision of the Faculty of Agriculture and Animal Production of King Saud University. All ewes were raised under the same environmental conditions for feeding, natural insemination, gestation, and parturition. The ewes were seasonally in lactation from December to April. They weighed an average of 61.7 ± 0.96 kg and had an average milk yield of 0.685 ± 0.347 mL/day at day 30 and 0.720 ± 0.170 mL/day at day 60. The animals received a total mixed ration (TMR consisting of alfalfa hay, corn, barley, soybean, molasse, minerals, and vitamins) without additives as shown in (Table 1). Feeding occurred twice daily at 8 a.m. and 4 p.m. (ad libitum) in the range of 1600–1800 kg DM/head and the animals had free access to clean water. The feed provided meets the requirements of the animals at the various stages of production according to NRC (2007) [12]. Prior to collecting a milk sample, we tested it for mastitis using the California Mastitis Test (CMT) to ensure that the animals udders were healthy.

### 2.2. Treatments and Design

A body condition assessment of the Najdi ewes was carried out 6–7 days after parturition. Thereafter, the ewes were divided into four groups as follows: Group-1 with a very low BCS, 2.5 points (*n* = 18); Group-2 with a low BCS, 3 points (*n* = 35); Group-3 a medium BCS, 3.5 points (*n* = 55); and Group-4 a high BCS, 4 points (*n* = 18). Ewes with a BCS between 2.5 and 3.5 points classified in the BCS as cases with a body condition score greater than 3.5 points, based on the 0.5—point scale (ranging from 1: extremely thin to 5: obese). In addition, the BCS was determined at 30 and 60 days of lactation.

### 2.3. Body Condition Measurements

The body condition score (BCS) was measured by a specialist to determine the amount of muscle and fat stored subcutaneously in the lumbar region of Najdi ewes. The BCS was used as a scale from 1 (lean) to 5 (obese) according to Jefferies [13]. Briefly, BCS 1 (poor) denotes animals in which no fat was palpable and the amount of muscle between skin and bone was small. The ribs, spine, lumbar vertebrae, and pelvic bones were prominent. BCS 2 (thin) represents animals that had a light layer of fat on the bones, but whose ribs were clearly visible. BCS 3 (intermediate) denotes animals with an overall smooth appearance and a light layer of fat over the ribs and apophysis of the lumbar vertebrae. With slight pressure, the ribs and lumbar vertebrae could be felt. BCS 4 (excess) denotes animals with a plump appearance and a visible layer of fat. The bone structures under the skin were palpated with moderate to firm pressure. BCS 5 (obesity) refers to animals with excessive body fat coverage and a round shape, and bony prominences could not be felt.

### 2.4. Collection and Analysis of Milk Samples

All ewes were milked at 8 a.m. after their lambs had been isolated for 12 hours. Samples were taken twice on days 30 and 60 of lactation. Samples (50 mL) were taken from the whole milk and analyzed with the Milko-Scan FT6000 (Foss, Hillerd, Denmark) to determine the milk components (fat, protein, lactose and total solids). Analysis of the fatty acid profile was conducted using gas chromatography–mass spectrometry (GC-MS). The fat extraction from 20 mL of the milk sample was stored at 20 °C until FA profiles analysis, according to Luna et al. [14]. FA methyl esters (FAMEs) were prepared, according to [15,16] and using Hexan. Mass spectrometry data were acquired and processed using the GC-MS Chem-Station data system. The proportion of each FA was determined from the ratio of the peak area of each FA to the total peak area of all FAs in the fat sample.

### 2.5. Statistical Analysis

The experimental data were analyzed through a completely randomized design, by two-way ANOVA (analysis of variance) and general linear model of procedure (Proc GLM), using the SAS (version 9.4, SAS Institute Inc., Cary, NC, USA). The statistical model was Y^ijK^ = µ+ BCS^i^ + LS_j_+ ε_ijK_, where: Y_ijK_ = milk composition and fatty acid profile; BCS_i_ = independent variable effect of the body condition score; LS_j_ = independent variable effect of the lactation stage on days 30 and 60 of lactation; and ε_ij_ = random error. Differences between means were compared using Duncan’s test (least statistical differences) considering *p* ≤ 0.05. Correlation analysis for 43 variables of milk composition and fatty acid profile with postpartum BCS was analyzed using Pearson’s correlation coefficient by XLstat -2023 statistical software [17].

## 3. Results

The results of the analytical studies are summarized in terms of milk composition on days 30 and 60 of lactation and related to the body condition scores of the Najdi ewes as shown in Table 2.

Based on the results of this study, the ewes with a high BCS of 3.5 and 4 showed a slight increase in milk fat percentage on day 30 of lactation. In contrast, milk protein percentage on days 30 and 60 of lactation was significantly higher (*p* ≤ 0.05) in ewes with medium body condition 3.5 while ewes with high body condition 4 were significantly lower (*p* > 0.05), as shown in Table 2. The difference in body condition between the experimental groups persisted throughout the experiment (*p* ≤ 0.05), and the effect of the interaction between the lactation stage and body condition was not significant.

In this study, BCS had significant effects on individual FAs on days 30 and 60 of lactation as shown in Table 3 and Table 4. The evaluation of a ewe’s milk with a high BCS of 4 on day 30 of lactation showed a significantly higher effect (*p* < 0.05) on the proportion of pentadecanoic acid (C15:0), palmitic acid (C16:0), palmitoleic acid (C16:1 cis 7), and (C21:0). In contrast, the ewes with a BCS of 2.5 had a significantly lower effect (*p* < 0.05) on stearic acid (C18:0), C18:2 cis 13 and cis 14 and LA (C18:2-n6) as shown in Table 3. 

On the other hand, at day 60 of lactation the medium BCS of 3 and 3.5 showed a significantly high effect (*p* < 0.05) on the proportion of small chai fatty acid (SCFA) including caproic acid (C6:0), caprylic acid (C8:0), capric acid (C10:0), lauric acid (C12:0), and myristic acid(C14:0). In contrast, in the ewes with high BCS of 4, a significantly higher level (*p* < 0.05) of odd chain fatty acids (C15:0 iso, C15:0 antiso, C17:0 iso, and C17:0), palmitoleic acid (C16:1 cis 7), and long chain fatty acids (LSFA) including stearic acid (C18:0), arachidic acid(C20:0), heneicosanoic acid(C21:0), pehenic acid(C22:0), arachidonic acid(C20:4), and adrenic acid (C20:4) was observed at day 60 of lactation, as shown in Table 4. It is worth noting that on day 60 of lactation, the levels of essential fatty acids (EFA) such as oleic acid (OA-C18:1 cis 9), vaccenic acid (VA-C18:1 cis 11), and LA (C18:2) of milk ewes with a poor BCS of 2.5 reached the highest value. 

The results in Table 5 show the effect of the BCS on total fatty acids (g/100 g FA) on days 30 and 60 of lactation. At day 60 of lactation, USFA and MUFA levels of milk in ewes with a BCS of 2.5 increased significantly (*p* < 0.01) by 9.05% and 8.93%, respectively, compared to other BCSs. By contrast, the SFA levels of milk in ewes with a BCS of 2.5 showed a significant decrease (*p* < 0.05) by (−3.70%) on day 60 of lactation, as shown in Table 5. The outcome analysis of OCFA is presented in Table 4 and Table 5, demonstrating that ewes with a BCS of 4 at day 60 of lactation had a significantly higher (*p* < 0.05) level of total OCFA as well as most OCFA such as C15:0 iso, C15:0 antiso, C15:0, C17:0 and C17:0 iso. 

The analysis of the correlation coefficients of BCS with milk components and fatty acid profiles are shown in Figure 1. A postpartum BCS was positively correlated with milk fat (*p* < 0.05; 0.049) and margaric acid C17:0 (0.023), while it was negatively correlated with LA C18:2 (*p* < 0.05; −0.002), CLA (*p* < 0.05; −0.03), and arachidonic acid C20:4 (*p* < 0.05; −0.01).

## 4. Discussion

After parturition, the physical condition of dairy animals deteriorates as they attempt to adapt physiologically to the increased energy demands of milk production; by altering their metabolism, they provide their lambs with necessary nutrients and thus ensure the survival and continued existence of the species [18,19]. Furthermore, the deterioration in the body condition of ewes in the first week after parturition would not be solely due to the onset of milk production but to an increase in milk production. However, milk production in sheep during lactation peaks four weeks after birth [3]. The aim of this study was to determine the influence of BCS and the lactation stage at days 30 and 60 on milk composition and FA profiles in Najdi sheep.

This study showed that the BCS and lactation stage had significant effects on milk composition and fatty acid profile. The protein content of the milk from ewes with a BCS of 3 and 3.5 was high compared to ewes with a BCS of 2.5 on days 30 and 60 of lactation. These results were similar to those reported by Pulina et al. [20] in Sarda dairy sheep and [21] in goats with improved BCSs that had a high milk protein content. Furthermore, in this study, BCS significantly correlated with milk fat, UFA, and CLA. It is noteworthy that the fat and lactose content of the milk was not affected by the state of the body, and no significant differences were found in any of the groups during the lactation stage, indicating that the energy balance of the ewes was not affected and could meet their requirements from their feed. This was consistent with the report by Pulina et al. [20] on Sarda dairy sheep.

In contrast, the levels of UFA, oleic acid C18:1 cis-9, VA C18:1 cis-11, and LA C18:2 cis-9-trans12 were higher in milk from ewes with a BCS of 2.5 at day 60 of lactation. In addition, long chain fatty acid (LCFA) levels such as stearic acid (C18:0) and oleic acid (C18:1 cis9) were high in milk in ewes with a BCS of 2.5. This can be explained by the fact that sheep adipose tissue undergoes mobilization and therefore produces oleic, palmitic, and stearic acid, especially when the ewes are in an NEB status [22,23]. Loften et al. [24] found that palmitic acid and oleic acid increase in blood levels after birth, while stearic acid decreases due to secretion into milk, and about 50% of oleic acid in the mammary gland is desaturated to stearic acid.

The Najdi ewes with medium and high BCS scores of 3, 3.5, and 4 had milk with the highest SCFA and LSFA levels and the lowest UFA levels at day 60 of lactation. In this study, the milk quality of sheep with a postpartum body score of 2.5 was observed to contain elevated UFA and MUFA levels. These can be attributed to the activity of the enzyme 9-desaturase in the mammary gland, resulting in a preferential milk fatty acid profile, particularly MUFA. As reported by Bernard et al. [25], the enzyme 9-desaturase in the mammary gland plays a key role in the synthesis of MUFA in milk by introducing a cis double bond between carbons 9 and 10 of FA, and is found only in ruminant products. This activity is important in influencing the FA profile and determining the nutritional quality of milk.

However, the side effects of a reduced BCS in sheep after birth was associated with ketosis, metabolic diseases, a decrease in milk production, impaired reproductive performance, and early embryonic death [26]. Although there are a few studies demonstrating the association between body condition and fatty acids in sheep’s milk, this study is the first to examine this in Najdi sheep. Therefore, this topic is interesting for our study.

The lactation stage showed effects on milk composition, with lower milk fat and higher protein content on day 60 of lactation. These results are consistent with the report by Dimitar et al. [27] in the Rhodope Tsigai sheep breed and [28] in Araucana Creole ewes, in which protein content increased as lactation progressed.

On the one hand, on day 60 of lactation, the levels of short chain fatty acid (SCFA) and ALA decreased slightly, while stearic acid and oleic acid levels increased. On the other hand, SFA levels significantly decreased by day 60 of lactation, while USF and MUFA levels significantly increased by day 60 of lactation. However, PUFA and OCFA levels remained stable during the lactation stage. These results were consistent with those reported by Sinanoglou et al. [29] for the Karagouniko and Chios sheep breeds and Strzałkowska et al. [30] in the Polish white goat.

Wilson et al. [31] reported that a large proportion of carbon atoms (C) in milk fat (0.43–0.54%) comes from body tissues, especially in the early stages of lactation, resulting in weight loss. This is based on the fact that C and the fatty acids in milk fat come from plasma lipids, which are of endogenous and dietary origin. Thus, almost 80% of the body’s C in milk fat is contained in the fatty acid.

We considered the increase of OCFA levels of milk fat in obese ewes with BCS 4 at day 60 of lactation. The milk contains other FAs, which are in the minority, but are of great importance. Among them, fatty acids with an odd number of carbon atoms and a branched chain (OCFA) are present, as well as unsaturated fatty acids with double bond and trans. They all have one connection in common: their origin lies largely in the fermenter, which is synthesized in the rumen by the enzymes of the microorganisms via the rumen processes of the feed [32]. The OCFA accounts for 2% of the total FA, with 15:0 and 17:0 being the most common and representative ratios [33].

Odd-chain FAs (C15:0, C17:0) found in mammalian milk and not synthesized by humans bind to metabolic regulators and act as factors in anemia, dyslipidemia, type 2 diabetes mellitus, inflammation, fibrosis, Alzheimer’s disease, etc. Studies suggest that they could be considered essential fatty acids [34].

In general, the lactation stage affected most individual FAs indirectly via fat mobilization equilibrium or in the mammary glands via de novo synthesis of FAs. On day 60 of lactation, the levels of long-chain FAs such as stearic acid, palmitoleic acid, and oleic acid increased in the milk ewes under NEB. This may be due to the uptake of non-esterified FAs from body fat mobilization or, to a lesser extent, lipid uptake [35]. On the other hand, in dairy sheep, short-chain fatty acids decreased with increased synthesis of long-chain FAs, and these FAs are synthesized from rumen acetate and are related to the quality and flavor of cheese [36].

## 5. Conclusions

The body condition score of the ewes after parturition and the lactation stage influenced the milk quality and components of Najdi sheep. Based on these results, it was shown that ewes with a BCS of 2.5 have an indirect impact on milk quality by increasing MUFA, UFA, and essential fatty acids such as oleic acid, vaccenic acid, and linolenic acid. In addition, day 60 of lactation was the preferred time for obtaining good milk with high MUFA and OCFA content and therefore healthy dairy products.

It appeared that Najdi dairy sheep had the ability to quickly adapt to surrounding environmental conditions, since the composition of their milk was not affected by the weak condition of their bodies after parturition.

## Figures and Tables

**Figure 1 vetsci-10-00552-f001:**
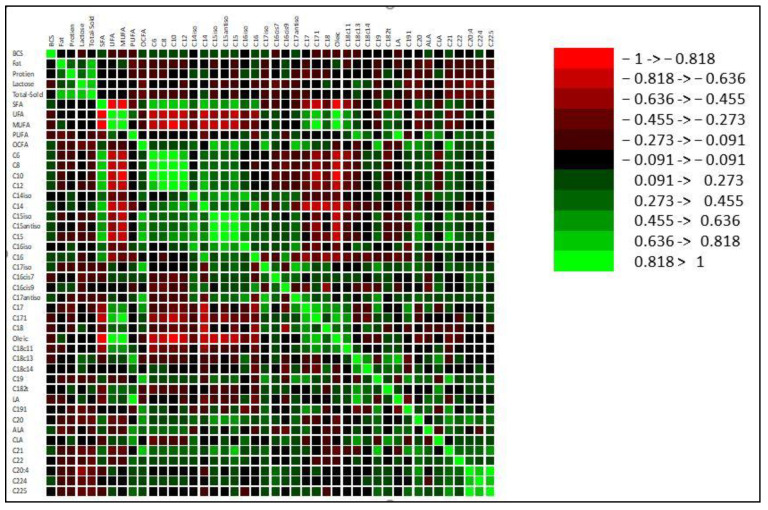
Correlation coefficient showing the relationships between the BCS with milk components and the fatty acid profile in the milk of Najdi sheep.

**Table 1 vetsci-10-00552-t001:** Chemical composition and fatty acid profile of total mixed ration (TMR) on a dry matter basis.

Nutrition’s	TMR
Chemical composition	
Dry matter%	89.96
Crude protein%	13.02
ME, Mcal/kg	2.87
NDF%	37.3
ADF%	24.1
Ash%	12.06
Fat%	2.43
Fatty acids profile (g/100g)	
C6:0	-
C8:0	0.12
C12:0	-
C14:0	0.12
C16:0	15.04
C16:1 cis 9	0.18
C17:0	0.12
C18:0	2.29
C18:1 trans 11	1.29
C18:1 cis 9	23.70
C18:2 cis 9, 12	51.43
C20:0	0.39
C18:3 cis 9, 12, 15	4.93
C22:0	0.29
C20:4 cis 7, 10, 13, 16	0.10

TMR: total mixed ration; NDF: neutral detergent fiber; ADF: acid detergent fiber; -: not detected.

**Table 2 vetsci-10-00552-t002:** Influence of the postpartum body condition score (BCS) on the milk composition of Najdi ewes at day 60 of lactation under a stable feeding system.

Components	Lactation Stage at Day 30				SEM	*p* Value
	BCS-2.5	BCS-3	BCS-3.5	BCS-4		
Fat	3.41	3.43	3.52	3.86	0.38	0.75
Protein	4.05 ^c^	4.26 ^ab^	4.37 ^a^	4.10 ^b^	0.12	0.051
Lactose	4.86	5.04	4.84	4.62	0.16	0.14
TS	13.08	13.03	13.21	13.38	0.44	0.89
	**Lactation Stage at day 60**					
Fat	3.41	3.25	3.52	3.33	0.27	0.74
Protein	4.60 ^b^	4.79 ^ab^	4.86 ^a^	4.14 ^c^	0.17	0.01
Lactose	5.27	4.65	4.95	4.50	0.36	0.23
TS	13.97	13.31	13.98	12.36	0.54	0.11

^a, b, c^: means with different superscripts within rows differed significantly at *p* < 0.05; SEM = Standard error of means; TS: total score.

**Table 3 vetsci-10-00552-t003:** Influence of the postpartum body condition score (BCS) on the fatty acid profile (g/100 g FA) of milk fat of Najdi ewes at day 30 of lactation under a stable feeding system.

Fatty Acids	Lactation Stage at Day 30				SEM	*p* Value
	BCS-2.5	BCS-3	BCS-3.5	BCS-4		
C6:0	1.25	1.19	1.24	1.31	0.06	0.39
C8:0	1.62	1.63	1.67	1.71	0.10	0.87
C10:0	5.79	5.58	5.59	5.76	0.25	0.92
C12:0	3.68	3.81	3.71	3.76	0.37	0.95
C14:0 iso	0.10	0.11	0.11	0.12	0.01	0.31
C14:0	8.32	9.42	9.17	9.62	0.39	0.06
C15:0 iso	0.27	0.30	0.31	0.32	0.02	0.35
C15:0 antiso	0.47	0.48	0.49	0.53	0.03	0.43
C15:0	0.94 ^c^	1.00 ^b^	1.02 ^ab^	1.11 ^a^	0.05	0.04
C16:0 iso	0.33	0.32	0.34	0.33	0.02	0.90
C16:0	24.32 ^c^	26.60 ^b^	26.95 ^ab^	27.31 ^a^	0.69	0.03
C16:1 cis 7	0.31	0.29	0.28	0.29	0.01	0.26
C16:1 cis 9	0.62 ^c^	0.67 ^b^	0.66 ^b^	0.77 ^a^	0.03	0.006
C17:0 iso	0.51	0.52	0.53	0.54	0.02	0.86
C17:0 antiso	0.71	0.69	0.71	0.70	0.02	0.95
C17:0	1.04	1.04	1.08	1.06	0.04	0.78
C17:1 cis 7	0.30	0.30	0.29	0.32	0.02	0.76
C18:0	14.31 ^a^	12.83 ^b^	13.01 ^b^	12.31 ^c^	0.51	0.03
C18:1 cis 9	26.71	24.83	24.39	24.34	1.09	0.28
C18:1 cis 11	0.50	0.45	0.46	0.47	0.02	0.21
C18:1 cis 13	0.36 ^a^	0.30 ^b^	0.30 ^b^	0.28 ^c^	0.02	0.02
C18:1 cis 14	0.35 ^a^	0.30 ^b^	0.29 ^b^	0.26 ^c^	0.02	0.003
C19:0	0.16	0.16	0.15	0.15	0.01	0.58
C18:2 (LA)	4.52 ^a^	3.91 ^b^	3.82 ^ab^	3.63 ^c^	0.19	0.003
C18:3 (ALA)	0.90 ^c^	0.95 ^b^	1.05 ^a^	0.91 ^c^	0.06	0.04
C18:3 (CLA)	071	0.76	0.71	0.73	0.04	0.59
C20:0	0.29	0.31	0.29	0.31	0.02	0.54
C21:0	0.07 ^c^	0.10 ^b^	0.10 ^b^	0.11 ^a^	0.01	0.03
C22:0	0.15	0.15	0.19	0.15	0.03	0.32
C20:4	0.31	0.36	0.31	0.31	0.04	0.36
C22:4	0.04	0.05	0.05	0.05	0.01	0.37
C22:5	0.15	0.19	0.18	0.17	0.02	0.44

^a, b, c^: means with different superscripts within row differed significantly at *p* < 0.05; LA: linolenic acid; ALA: α-linoleic acid; CLA: conjugated fatty acid; SEM = Standard error of means.

**Table 4 vetsci-10-00552-t004:** Influence of the postpartum body condition score (BCS) on the fatty acid profile (g/100 g FA) of milk fat of Najdi ewes at day 60 of lactation under a stable feeding system.

Fatty Acids	Lactation Stage at Day 60				SEM	*p* Value
	BCS-2.5	BCS-3	BCS-3.5	BCS-4		
C6:0	0.81 ^c^	1.04 ^a^	0.97 ^b^	0.98 ^b^	0.05	0.001
C8:0	0.95 ^c^	1.34 ^a^	1.23 ^ab^	1.20 ^b^	0.07	0.003
C10:0	3.22 ^c^	4.64 ^a^	4.29 ^ab^	4.07 ^b^	0.30	0.002
C12:0	2.24 ^c^	3.07 ^a^	2.93 ^ab^	2.67 ^b^	0.19	0.003
C14:0 iso	0.12	0.13	0.13	0.12	0.01	0.74
C14:0	8.21 ^c^	8.93 ^ab^	9.14 ^a^	8.24 ^b^	0.30	0.02
C15:0 iso	0.24 ^c^	0.29 ^b^	0.29 ^b^	0.32 ^a^	0.01	0.01
C15:0 antiso	0.42 ^c^	0.51 ^b^	0.50 ^b^	0.52 ^a^	0.03	0.02
C15:0	0.84	0.94	0.93	0.99	0.04	0.08
C16:0 iso	0.33	0.35	0.36	0.34	0.02	0.25
C16:0	26.15	26.95	27.02	26.55	0.60	0.46
C16:1 cis 7	0.32 ^b^	0.32 ^b^	0.30 ^c^	0.34 ^a^	0.01	0.03
C16:1 cis 9	0.78	0.75	0.76	0.75	0.03	0.91
C17:0 iso	0.53 ^b^	0.51 ^c^	0.52 ^b^	0.57 ^a^	0.01	0.01
C17:0 antiso	0.70	0.69	0.70	0.75	0.02	0.06
C17:0	1.08 ^a^	0.95 ^c^	0.98 ^b^	1.05 ^a^	0.03	0.01
C17:1 cis 7	0.37 ^a^	0.30 ^c^	0.31 ^b^	0.31 ^b^	0.01	0.02
C18:0	14.32 ^ab^	14.02 ^b^	13.56 ^c^	15.05 ^a^	0.42	0.050
C18:1 cis 9	30.33 ^a^	26.45 ^c^	27.17 ^b^	27.27 ^b^	0.95	0.001
C18:1 cis 11	0.57 ^a^	0.52 ^b^	0.52 ^b^	0.50 ^c^	0.02	0.02
C18:1 cis 13	0.32	0.29	0.31	0.31	0.01	0.29
C18:1 cis 14	0.28 ^c^	0.30 ^b^	0.29 ^b^	0.34 ^a^	0.01	0.03
C19:0	0.14 ^b^	0.13 ^c^	0.13 ^c^	0.16 ^a^	0.01	0.02
C18:2 (LA)	4.09 ^a^	3.73 ^c^	3.93 ^b^	3.75 ^c^	0.14	0.04
C18:3 (ALA)	0.66	0.64	0.68	0.64	0.02	0.17
C18:3 (CLA)	0.77	0.79	0.80	0.73	0.03	0.32
C20:0	0.27 ^c^	0.32 ^ab^	0.31 ^b^	0.33 ^a^	0.01	0.04
C21:0	0.05 ^c^	0.07 ^a^	0.06 ^b^	0.07 ^a^	0.01	0.02
C22:0	0.09 ^c^	0.12 ^a^	0.11 ^b^	0.13 ^a^	0.01	0.02
C20:4	0.30 ^c^	0.39 ^a^	0.33 ^b^	0.38 ^a^	0.02	0.02
C22:4	0.03 ^c^	0.05 ^ab^	0.04 ^b^	0.07 ^a^	0.01	0.001
C22:5	0.15	0.19	0.17	0.18	0.01	0.10

^a, b, c^: means with different superscripts within row differed significantly at *p* < 0.05; LA: linolenic acid; ALA: α-linoleic acid; CLA: conjugated fatty acid; SEM = Standard error of means.

**Table 5 vetsci-10-00552-t005:** Influence of the postpartum body condition score (BCS) on the total fatty acid profile (g/100 g FA) of milk fat of Najdi ewes at 30 and 60 days of lactation under a stable feeding system.

Total FA%	Lactation Stage at Day 30				SEM	*p* Value
	BCS-2.5	BCS-3	BCS-3.5	BCS-4		
SFA	63.87	66.25	66.58	67.18	1.19	0.16
UFA	36.13	33.75	33.42	32.82	1.19	0.16
MUFA	29.23	27.25	26.78	26.82	1.13	0.28
PUFA	6.82	6.44	6.59	6.01	0.26	0.06
OCFA	4.15	4.30	4.36	4.50	0.14	0.26
	**Lactation Stage at day 60**					
SFA	60.71 ^c^	65.01 ^a^	64.12 ^b^	64.11 ^b^	1.02	0.001
UFA	39.29 ^a^	34.99 ^c^	35.88 ^b^	35.89 ^b^	1.02	0.001
MUFA	33.05 ^a^	28.98 ^c^	29.70 ^b^	29.90 ^b^	0.81	0.001
PUFA	6.24	6.01	6.18	5.98	0.15	0.43
OCFA	4.01 ^c^	4.10 ^b^	4.11 ^b^	4.44 ^a^	0.12	0.04

^a, b, c^: means with different superscripts within rows differed significantly at *p* < 0.05; SEM = Standard error of means; SFA: saturated fatty acid; UFA: unsaturated fatty acid; MUFA: monounsaturated fatty acid; PUFA: polyunsaturated fatty acid; OCFA: Odd-chain fatty acids.

## Data Availability

Informed consent was obtained from all subjects involved in the study.

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
