# Peer review of "Postpartum Body Condition Score (BCS) and Lactation Stage (30 and 60 Days) Affecting Essential Fatty Acids (EFA) and Milk Quality of Najdi Sheep"

_vetsci, 2023, doi:10.3390/vetsci10090552_

Round 1

Reviewer 1 Report

The publication provides further information on the factors affecting milk yield in sheep

The page may be published after taking into account the comments addressed to the authors of the work.

Comments

  1.  The abstract is written correctly
  2. The introduction needs a major rewrite. Authors should refer citations to sheep and not to cattle. Many factors affect the milk yield of sheep. Cattle lactation cannot be directly related to sheep lactation. The biological conditions of sheep are completely different than in cattle
  3. Material and methods should be added: age of sheep, fecundity of mothers (how many lambs they gave birth), whether they are seasonal or aseasonal sheep. What time of year do sheep give birth to lambs? Nutrition should be described in detail. Were the sheep brewed before mating? What was the condition during the mating period?
  4. Results described correctly,
  5. The discussion should be reworded and more attention should be paid to the comparison of the obtained results to the literature in the field of sheep and goats, and not to the performance of cattle. Metabolic, physiological and hormonal changes, fatty acid synthesis are species-specific (sheep's milk has a specific composition) and should not be compared to cow's milk
  6. The conclusion should contain information on the application of the results to practice

Author Response

Dear Prof,

Thank you for giving us the opportunity to submit a revised draft of our manuscript titled Postpartum body condition score (BCS) and lactation stage (30 and 60 days) affecting essential fatty acids (EFA) and milk quality of Najdi sheepto the Veterinary Sciences. We appreciate the time and effort that you and the reviewers have dedicated to providing your valuable feedback on our manuscript. We are grateful to the reviewers for their insightful comments on our paper. We have been able to incorporate changes to reflect most of the suggestions provided by the reviewers. We have highlighted the changes within the manuscript. Based on this, we are optimistic that the revised version of the manuscript would achieve certain publishable status.

Reviewer 2 Report

The research topic is interesting and current, but the manuscript is unfortunately very poorly written for several reasons: a) it is written in very poor English so some sentences and parts of the manuscript are difficult to understand; b) the manuscript is not written systematically (for example, in the Results chapter, the authors first state the results about fatty acids in milk, then go to the chemical composition of milk, and then again about individual fatty acids); c) you do not use the correct and usual professional terminology (eg they use the term "calving" for lambing sheep, then they use the terms peak of lactation and stage of lactation (which are not the same)...).

In addition, from the Materials and methods, it is not clear to the reader how the research was carried out, for example, what does it mean: 30 days peak of lactation and 60 days peak of lactation - are the compared sheep that achieved the peak of lactation production on the 30th day after lambing, i.e. sheep that achieved peak lactation milk production on the 60th day after lambing; or was a milk sample taken on the 30th and 60th day of lactation for each sheep, so we are talking about the influence of the stage of lactation, and not the peak of lactation milk yield? If the latter is true, then the term peak of lactation is neither necessary nor correct. That is why I think that the Material and methods should be explained in more detail. Also, it is not clear how the sheep are grouped into groups based on body condition score (lines 102-108). Vaguely written material and methods make it much more difficult for the reader to follow the results of the research. Also, in the Results, I think that Figures 1 and 2 are not necessary, since the same is shown in tables 3-5 (also, from graphs 1 and 2, the values of statistical significance of the differences between individual groups of sheep are not evident, and the authors refer to them at the beginning of the chapter The results.

The list of used references could be somewhat more current (please use more references published in last 5 years).

The conclusion begins with the sentence: "The body condition score of ewes at birth and in peak lactation affected the milk 312 quality and milk components of Najdi sheep" - does this mean that you assessed the body condition twice on each sheep, or a few days after birth as you stated in Materials and methods?

The manuscript is written in very poor English so some sentences and parts of the manuscript are difficult to understand. Also, authors didn't use the correct and usual professional terminology.

Author Response

(The authors gave the same response as above.)

Round 2

Reviewer 2 Report

I would be more satisfied and the article would have much more scientific weight if the influence of the stage of lactation on the chemical composition of milk and the content of fatty acids was shown in relation to the condition of the sheep (four groups of sheep with regard to the evaluation of the condition of the sheep and for each of them the content of milk constituents and of fatty acids in each stage of lactation; 30 and 60 days per parturition). In this way, the movement of the content of individual components of milk and fatty acids according to the researched stages of lactation would be evident). As it is, the paper is quite average and correctly written, and I leave the final decision on acceptance of the manuscript to the Editorial Board.

The quality of the English language is significantly better in this corrected version of the manuscript. Professional terminology is also more understandable and accurate compared to the first version of the manuscript.

Author Response

We thank the editor and reviewer for providing us with some valuable and helpful comments that can be used to improve our manuscript to attain a publishable standard. Based on these suggestions and comments, we have revised the manuscript thoroughly. 
•    We edited all reviewer comments and suggestions by edited statistical analysis, changing tables, results, and conclusion
•    All changes in the manuscript were made and highlighted with yellow color. 

-    We greatly appreciate the reviewer’s comments and suggestions, which have contributed significantly to what we believe is now a much-improved paper and of sufficiently strong standard.